# GEOMETRY-AWARE IMAGE FLOW MATCHING

## ABSTRACT

Recent advances in image generation, including diffusion models and flow matching, have achieved remarkable success through mathematical foundations. Furthermore, when the underlying data manifold is known, geometry-aware generative models leveraging differential geometric tools have demonstrated superior performance by exploiting intrinsic geometric structure. However, natural images lack explicit geometric priors, forcing existing methods to operate solely in high-dimensional Euclidean space despite potential geometric constraints in the data. In this work, we investigate the underlying geometric structure of natural images and introduce geometry-aware image flow matching methods. Through directional decomposition analysis, we observe that the majority of semantic information in images is encoded in their directional components, while scalar components can be effectively approximated by global average of dataset with minimal impact on quality. This property appears not only in RGB space, but also extends to various latent spaces, indicating that natural images can be generally projected on a hypersphere. Building on this finding, we introduce *geometry-aware image flow matching*: Spherical Optimal Transport Flow Matching (SOT-CFM), which leverages angular distance metrics, and Spherical Flow Matching (SFM), which constrains dynamic directly on the hypersphere. Experiments on CIFAR-10 and ImageNet confirm that our spherical methods achieve competitive or superior performance compared to their Euclidean counterparts, paving the way for future advances in *geometry-aware* image generative modeling.

## 1 INTRODUCTION

Image generation has seen rapid progress through successive paradigms, from Continuous Normalizing Flows (CNF) (Chen et al., 2018; Grathwohl et al., 2018) to Diffusion models (DM) (Song & Ermon, 2019; Sohl-Dickstein et al., 2015; Ho et al., 2020; Song et al., 2021; Dhariwal & Nichol, 2021; Karras et al., 2022), and more recently to Flow Matching (FM) (Lipman et al., 2023; Liu et al., 2022; Albergo et al., 2023) approaches. Each breakthrough has delivered increasingly impressive results in terms of sample quality, training stability, and generation efficiency. However, despite these advances, all these methods fundamentally rely on Euclidean geometry assumptions, treating images as vectors in high-dimensional Euclidean space. While this approach has proven successful, it may not fully capture the intrinsic geometric structure of natural images. If we could better understand and leverage the true geometry of image data, we might achieve more principled and effective generative modeling.

In domains where the underlying data manifold is known, geometry-aware generative modeling has delivered tangible gains. Early work on Riemannian CNF (Mathieu & Nickel, 2020) parameterizes flexible densities directly on smooth manifolds by integrating ODEs on the manifold. Subsequent Riemannian score-based (Bortoli et al., 2022) and Riemannian Diffusion models (Huang et al., 2022) generalized score estimation and diffusion samplers to arbitrary Riemannian manifolds via manifold-aware divergences. More recently, Riemannian Flow Matching (RFM)(Chen & Lipman, 2024) removed simulator bias by matching geodesic velocities with closed-form target vector fields. In application domains where geometry is dictated by symmetries (*e.g.*, periodic crystals), FlowMM(Miller et al., 2024) extends RFM with group-equivariant structure, reporting state-of-the-art structure generation with substantially fewer integration steps. Collectively, these methods exploit geodesics, parallel transport, and manifold-aware metrics to obtain higher-quality samples, faster convergence, and more principled training relative to Euclidean baselines when the

geometric prior is correct. However, a fundamental challenge remains: unlike structured domains with well-understood geometric priors, the intrinsic manifold structure of natural images is largely unknown. Images do not come with explicit geometric constraints or symmetries for defining a Riemannian manifold. This uncertainty has prevented the direct application of geometry-aware generative modeling to image domains, restricting work to Euclidean methods despite their potential limitations.

In this work, we propose a solution through directional decomposition analysis. We demonstrate that when images are decomposed into their directional and scalar components—treating each image vector as having both a direction (unit vector) and magnitude (norm)—the vast majority of semantic information is encoded in the directional component. Remarkably, we find that the scalar component can be approximated by simply using the dataset's average norm without significant loss of reconstruction quality. While this finding may appear intuitive for RGB spaces, it is far less evident in latent representations learned through reconstruction objectives. Through extensive experiments, we demonstrate that, even in complex latent spaces of diverse autoencoders, directional component serves as the primary factor for faithful image reconstruction, while the global average of scalar component suffices to retain most visual content. This observation suggests that natural images can be effectively regarded as data points lying on a hypersphere with a radius determined by the dataset's average magnitude, both in RGB and latent spaces.

This finding enables us to establish geometry-aware image flow matching by either projecting data onto hyperspheres to leverage spherical geometry or by utilizing directional metrics instead of Euclidean metrics between vectors. Beyond the geometric benefits, this spherical projection provides a strong training advantage by simplifying the learning task—since all projected data points reside on the same sphere with predetermined magnitude, models can focus exclusively on learning directional dynamics rather than jointly optimizing both direction and magnitude components. Leveraging these insights, we introduce two approaches that adapt existing flow matching methods to spherical geometry: Spherical Optimal Transport Conditional Flow Matching (SOT-CFM), which adapts OT-CFM (Tong et al., 2024; Pooladian et al., 2023) by employing angular distance metrics instead of Euclidean distances for optimal transport coupling, and Spherical Flow Matching (SFM), which operates entirely on the hyperspherical manifold by projecting both source and target distributions onto the sphere and using geodesic paths as the optimal transport trajectories between data points.

We validate our geometric approach through comprehensive experiments on CIFAR-10 and ImageNet-256, benchmarking our spherical methods against established Euclidean flow matching baselines such as Independent CFM (I-CFM) and Optimal Transport CFM (OT-CFM). The results confirm that our spherical projection reduces the cost of matching image vector magnitudes, SOT-CFM benefits from angular distance metrics, and most significantly, SFM achieves a fundamental breakthrough as the first successful application of Riemannian manifold-based generative methods to natural images, establishing a new paradigm that transforms image generation from treating images as arbitrary vectors to leveraging their intrinsic spherical geometry.

The key contributions of this work are threefold: (1) we reveal that natural images possess inherent spherical geometric structure through directional decomposition analysis, (2) we introduce two geometry-aware flow matching methods (SOT-CFM and SFM) that leverage this structure for improved generation, and (3) most importantly, we demonstrate the first successful application of manifold-based generative modeling to natural images, making sophisticated geometric tools accessible to image generation domains for the first time.

## 2 PRELIMINARY

**Conditional Flow Matching (CFM)** (Lipman et al., 2023) provides a simulation-free approach for training Continuous Normalizing Flows (CNFs) by learning a time-dependent vector field $u_t(x)$ : $[0, 1] \times \mathbb{R}^d \to \mathbb{R}^d$ that generates a flow from a source distribution $p_0$ to a target data distribution $p_1$. The flow is governed by the ordinary differential equation:

$$\frac{dx}{dt} = u_t(x_t), \quad x_0 \sim p_0. \tag{1}$$

Unlike traditional CNF training methods that require expensive simulation during training, Conditional Flow Matching optimizes the regression objective:

$$\mathcal{L}_{\text{CFM}}(\theta) = \mathbb{E}_{t,x_0,x_1,x_t} \left[ \|v_\theta(t, x_t) - u_t(x_t|x_0, x_1)\|^2 \right], \tag{2}$$

where $u_t(x_t|x_0, x_1)$ is the conditional vector field, $x_t$ is sampled from the conditional probability path $p_t(x|x_0, x_1)$, and $v_\theta$ is the learned vector field parameterized by $\theta$.

**Optimal Transport CFM (OT-CFM).** Standard Conditonal Flow Matching uses independent coupling between source and target distributions, known as Independent Conditional Flow Matching (I-CFM), which can result in inefficient transport paths. Optimal Transport Conditional Flow Matching (OT-CFM) (Tong et al., 2024; Pooladian et al., 2023) addresses this by finding optimal pairings between source and target points using optimal transport theory.

Instead of independent sampling from $p_0$ and $p_1$, OT-CFM solves the optimal transport problem:

$$\min_{\pi \in \Pi(p_0, p_1)} \mathbb{E}_{(x_0, x_1) \sim \pi}[c(x_0, x_1)], \tag{3}$$

where $c(x_0, x_1)$ is a cost function (typically $\|x_0 - x_1\|^2$) and $\Pi(p_0, p_1)$ denotes the set of all joint distributions with marginals $p_0$ and $p_1$. In practice, the exact optimal coupling $\pi$ cannot be computed for continuous distributions, so mini-batch optimal transport approximation is employed, where the optimal coupling is computed only over finite mini-batches. This approach creates simpler flows with straighter trajectories that are more stable to train and enable faster inference.

**Riemannian Flow Matching (RFM).** While standard Flow Matching operates in Euclidean space, many applications benefit from incorporating geometric structure. Riemannian Flow Matching (RFM) (Chen & Lipman, 2024) extends flow matching to Riemannian manifolds $\mathcal{M}$ equipped with a metric tensor $g$. On a Riemannian manifold, the flow evolves according to:

$$\frac{dx}{dt} = u_t(x_t), \quad x_t \in \mathcal{M} \tag{4}$$

where $u_t(x) \in T_x\mathcal{M}$ is a time-dependent vector field in the tangent space at $x$.

The key innovation of RFM is constructing conditional vector fields using geodesics, the shortest paths on the manifold. For a conditional flow from $x_0$ to $x_1$, the conditional vector field is defined as

$$u_t(x_t|x_0, x_1) = \frac{d}{dt}\gamma_t(x_0, x_1)\Big|_{s=t}, \tag{5}$$

where $\gamma_t(x_0, x_1)$ is the geodesic connecting $x_0$ and $x_1$, parameterized by $t \in [0, 1]$.

The RFM training objective is given by

$$\mathcal{L}_{\text{RFM}}(\theta) = \mathbb{E}_{t,x_0,x_1,x_t} \left[ \|v_\theta(t, x_t) - u_t(x_t|x_0, x_1)\|_g^2 \right], \tag{6}$$

where $\|\cdot\|_g$ denotes the norm induced by the Riemannian metric $g$.

## 3 FLOW MATCHING ON SPHERICAL GEOMETRY

In Section 2, we outline the basic principles of Flow Matching and its extensions, showing that RFM can achieve superior performance when the underlying manifold structure is known. However, as noted in Section 1, the challenge for image generation lies in the absence of explicit geometric priors—natural images do not come with predefined manifold structure that can be directly exploited by existing geometry-aware methods.

In this section, we address this fundamental limitation by demonstrating how geometric structure can be discovered within image data itself. Our approach centers on a key insight: through analysis of directional and scalar decomposition, we reveal that images naturally exhibit affinity for spherical geometry. This discovery enables us to use *geometry-aware* approaches for image generation for the first time.

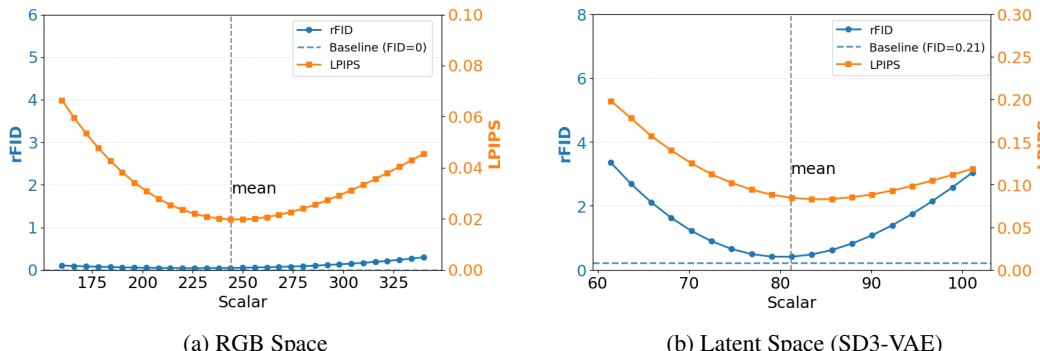

(a) RGB Space                    (b) Latent Space (SD3-VAE)

Figure 1: **Robustness analysis of spherical projection on ImageNet256.** We project the dataset onto hyperspheres with various scalar (radii) while preserving directional components, then evaluate reconstruction quality using rFID and LPIPS metrics. The vertical dashed line indicates mean scalar of the dataset. The results show near-baseline performance (rFID = 0 for RGB, rFID =0.41 for latent) across a wide range of values around the mean, demonstrating that spherical projection is not sensitive to scalar choice and confirming that directional information dominates semantic content in image data.

### 3.1 VECTOR DECOMPOSITION AND DIRECTIONAL ANALYSIS

To understand the geometric structure underlying image data, we treat each image as a flattened vector in $\mathbb{R}^d$ and begin with a basic mathematical observation: any such vector can be naturally decomposed into its directional and scalar components. Formally, given an image vector $x \in \mathbb{R}^d$, we can express it as:

$$x = \|x\|_2 \cdot \frac{x}{\|x\|_2} = s \cdot \hat{x}, \tag{7}$$

where $s = \|x\|_2$ is the magnitude (scalar component) and $\hat{x} = x/\|x\|_2$ is the unit direction vector lying on the $(d-1)$-dimensional unit hypersphere $\mathbb{S}^{d-1}$.

When applied to image data, we can separate each image vector into a direction and a scalar component. While the intrinsic manifold structure of image data is not explicitly known, the directional vectors naturally reside on the unit hypersphere. This geometric constraint suggests that if data points share similar magnitudes, images can be regarded as lying on a hypersphere of constant radius. However, we observe that image magnitudes span a broad range, which makes such an assumption invalid in practice. Nevertheless, if the data projected onto a common sphere with a fixed radius still preserves the semantic content of the images, then it becomes reasonable to treat the image manifold as spherical.

To explore this hypothesis, we project image datasets onto hyperspheres of varying radii in both RGB and various latent spaces, preserving only the directional components while modifying the scalar component. We then measure reconstruction quality using rFID and LPIPS metrics compared to the original dataset. As shown in Fig. 1a, rFID remains near zero across a wide range of radii in RGB space and LPIPS stays consistently low. Remarkably, as shown in Fig. 1b, we observe similar robustness patterns in SD3-VAE's latent space. Furthermore, we find that this phenomenon extends across multiple autoencoder latent spaces used in the LDM framework, demonstrating the generality of this finding (see Table 1).

These findings show that most of the meaningful information in images lies in the directional component, while the scalar component can be well-approximated by a global average. Based on this observation, we can project all data onto a single hypersphere, which offers significant advantages for generative modeling. First, by eliminating the need to match scalar components, the model can dedicate its entire capacity to learning the semantically important directional variations, reducing training complexity. Second, this spherical projection naturally enables *geometry-aware image generative modeling* by providing an explicit geometric structure to exploit.

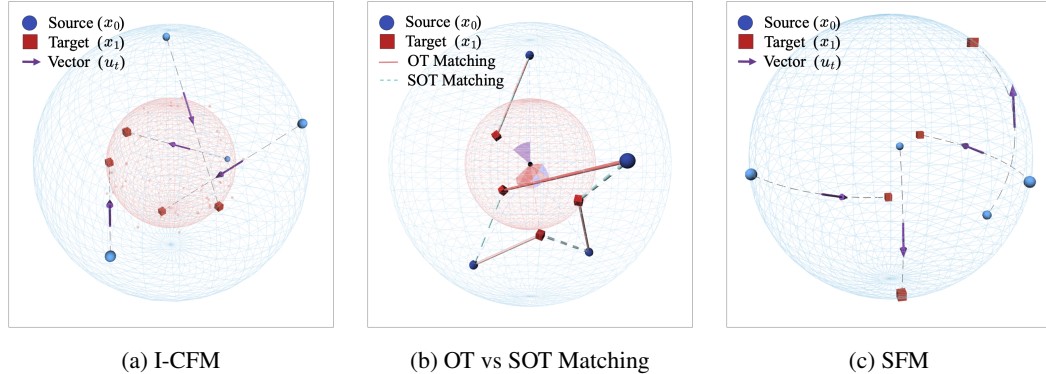

(a) I-CFM                  (b) OT vs SOT Matching                  (c) SFM

Figure 2: Comparison of different flow matching strategies. **(a) I-CFM:** Samples from a prior distribution ($x_0$, blue circles ●) are randomly paired with data samples ($x_1$, red squares ■) via straight-line paths in Euclidean space. **(b) OT vs. SOT Matching:** Standard OT (solid red —) minimizes Euclidean distance, potentially creating pairings with large angular differences. SOT (dashed cyan ---) matches points by angular proximity, better preserving semantic structure. **(c) SFM:** Flows are constrained to a spherical manifold with geodesic paths (great-circle arcs) between samples and tangent vector fields ($u_t$).

## 3.2 Spherical OT-CFM (SOT-CFM) with Angular Metrics

The observation that image semantics are primarily encoded in directional components provides a natural motivation to revisit OT-CFM (Tong et al., 2024; Pooladian et al., 2023) through the lens of spherical geometry. Recall that OT-CFM seeks an optimal coupling between the source distribution $p_0$ and target distribution $p_1$ by minimizing the expected transport cost in Eq. (3), where the cost $c(x_0, x_1)$ is typically the squared Euclidean distance $\|x_0 - x_1\|^2$. While this Euclidean formulation is effective for datasets without known geometric structure, our analysis suggests that the Euclidean formulation underrepresents the directional nature of image semantics. Since directional information dominates semantic content, the angular separation between images—rather than their Euclidean distance—provides a more meaningful measure of similarity. This insight suggests that transport costs should reflect the geometry of the underlying data manifold.

We therefore introduce Spherical OT-CFM (SOT-CFM), which replaces the Euclidean cost with a metric that operates directly on the directional components of the data. Specifically, we define the transport cost as the angular distance between two data points $x_0$ and $x_1$:

$$c_{\text{ang}}(x_0, x_1) = \arccos\left( \frac{\langle x_0, x_1 \rangle}{\|x_0\|_2 \|x_1\|_2} \right). \tag{8}$$

This angular cost function is invariant to the magnitudes of $x_0$ and $x_1$, ensuring that the optimal transport plan prioritizes matching images based on their core semantic content (direction) rather than their scalar attributes.

## 3.3 Spherical Flow Matching

While SOT-CFM addresses transport cost issues by replacing Euclidean distance with angular distance, the spherical nature of image data can be leveraged more directly. Rather than only modifying the coupling strategy, we propose Spherical Flow Matching (SFM), which constrains both source and target distributions to the hypersphere manifold $\mathbb{S}^{d-1}$ and defines flow paths as geodesics on the manifold, allowing the entire flow dynamics to operate within the spherical geometry.

This approach is well-motivated for generative modeling: in high dimensions, samples from a Gaussian distribution naturally concentrate near the surface of a hypersphere. This property allows us to project both the source Gaussian distribution and target image data onto the same hypersphere, enabling flow dynamics that operate entirely within the geometric space where semantic information resides.

On the hypersphere, the shortest path connecting any two points is the geodesic, which has a closed-form expression as spherical linear interpolation (slerp):

$$\tilde{x}_t = \gamma_t(\tilde{x}_0, \tilde{x}_1) = \frac{\sin((1-t)\theta)}{\sin\theta}\tilde{x}_0 + \frac{\sin(t\theta)}{\sin\theta}\tilde{x}_1, \quad t \in [0,1], \tag{9}$$

where $\tilde{x}_0 = r \cdot x_0/\|x_0\|_2$ and $\tilde{x}_1 = r \cdot x_1/\|x_1\|_2$ are the projected vectors on the hypersphere of radius $r$, and $\theta = \arccos(\langle \tilde{x}_0, \tilde{x}_1 \rangle / r^2)$ is the angle between them.

Following the geodesic path, we can derive the conditional vector field $u_t$ at any point $\tilde{x}_t$ along the trajectory. By construction, this vector field $u_t$ is always in the tangent space $T_{x_t}\mathbb{S}^{d-1}$ for all $t \in [0,1]$. Our goal is to train a model $v_\theta(t, x_t)$ to predict this tangent vector. Unlike Euclidean flow matching, SFM measures the discrepancy using the Riemannian inner product induced by the hypersphere geometry. Specifically, for a base point $x_t$ and tangent vectors $u, v \in T_{x_t}\mathbb{S}^{d-1}$, the inner product is:

$$\langle u, v \rangle_{g(\tilde{x}_t)} = u^\top v, \tag{10}$$

since the hypersphere inherits the Euclidean metric restricted to the tangent space. The SFM loss is thus formulated as:

$$\mathcal{L}_{\text{SFM}}(\theta) = \mathbb{E}_{t, \tilde{x}_0, \tilde{x}_1, \tilde{x}_t} \left[ \|v_\theta(t, \tilde{x}_t) - u_t(\tilde{x}_t|\tilde{x}_0, \tilde{x}_1)\|^2 \right]. \tag{11}$$

By optimizing this geometrically-grounded loss, SFM effectively constrains the entire generative process to the hypersphere, where crucial semantic information resides. This approach is a critical step forward as it successfully introduces manifold-based generative methods, which have previously been confined to theoretical or synthetic datasets, into the practical domain of natural image generation for the first time. Our work demonstrates that geometry-aware frameworks can break free from theoretical constraints to become viable tools for real-world image synthesis, establishing a foundation for future models that exploit the intrinsic geometric structure of natural data.

### 3.4 MAGNITUDE PREDICTION FOR FINE-GRAINED ADJUSTMENTS

While hyperspherical projection using the global average magnitude $\bar{s} = \frac{1}{N}\sum_{i=1}^{N}\|x_i\|_2$ for dataset $\{x_i\}_{i=1}^{N}$ effectively preserves essential information, it incurs a slight loss in reconstruction fidelity. To address this limitation, we introduce a lightweight Magnitude Refinement Network ($N_\phi$) for fine-grained adjustments. The network takes the unit direction vector $\hat{x}$ as input and predicts a magnitude correction $\Delta s_\phi$, which approximates the true deviation $\Delta s = s - \bar{s}$. The final refined vector is then constructed as

$$x_{\text{pred}} = (\bar{s} + \Delta s_\phi) \cdot \hat{x}. \tag{12}$$

The network $N_\phi$ is optimized with a composite loss that combines pixel-level reconstruction (MSE) and perceptual similarity (LPIPS). For models operating directly in image space (e.g., RGB), the objective is

$$\mathcal{L}_{\text{img}}(\phi) = \mathbb{E}_{x, x_{\text{pred}}} \left[ \|x - x_{\text{pred}}\|_2^2 + \lambda \mathcal{L}_{\text{LPIPS}}(x, x_{\text{pred}}) \right]. \tag{13}$$

For latent-space models, we apply the same refinement process to the unit latent vector $\hat{z}$, yielding $z_{\text{pred}} = (\bar{s} + \Delta s_\phi) \cdot \hat{z}$. The losses are computed after decoding through $\mathcal{D}$, with an additional latent consistency term:

$$\mathcal{L}_{\text{latent}}(\phi) = \mathbb{E}_{z, z_{\text{pred}}} \left[ \|x - \mathcal{D}(z_{\text{pred}})\|_2^2 + \lambda_1 \mathcal{L}_{\text{LPIPS}}(x, \mathcal{D}(z_{\text{pred}})) + \lambda_2 \|z - z_{\text{pred}}\|_2^2 \right]. \tag{14}$$

This refinement strategy enhances representations that are already well-approximated by the global average, further improving reconstruction fidelity while introducing negligible computational overhead at inference time.

## 4 EXPERIMENT

### 4.1 EXPERIMENTAL SETUP

**Datasets.** We conduct experiments on two standard image generation benchmarks: CIFAR-10 (Krizhevsky et al., 2009) and ImageNet-256 (Russakovsky et al., 2015). CIFAR-10 consists of

50,000 training images across 10 classes at $32 \times 32$ resolution, while ImageNet-256 contains approximately 1.28 million training images from 1,000 classes at $256 \times 256$ resolution. For ImageNet-256, we perform class-conditional generation to evaluate our methods' ability to incorporate semantic conditioning.

**Model Architecture.** For CIFAR-10, we employ a standard U-Net architecture (Dhariwal & Nichol, 2021) commonly used in diffusion and flow matching methods. For ImageNet-256, we adopt DC-AE (Chen et al., 2024) as the autoencoder for efficiency and DiT XL/1 (Peebles & Xie, 2023) as the diffusion model. And we adopt MobileNetV2 (Sandler et al., 2018) for magnitude prediction.

**Training Configuration.** For CIFAR-10, we train all models for 200,000 iterations with a batch size of 512, learning rate of $2 \times 10^{-4}$, and Adam optimizer (Kingma & Ba, 2014). Training is performed on a single NVIDIA A6000 GPU. For ImageNet-256, we train for 140,000 iterations with a batch size of 1,024, learning rate of $2 \times 10^{-4}$, and AdamW optimizer (Loshchilov & Hutter, 2017) with weight decay of 0 and $\beta = (0.9, 0.95)$. Training is conducted on 2 NVIDIA A100 40G GPUs.

**Baselines.** We compare our proposed methods against two fundamental flow matching approaches: Independent Conditional Flow Matching (I-CFM) and Optimal Transport Conditional Flow Matching (OT-CFM). These baselines represent the current standard for flow-based generative modeling and provide a fair comparison for evaluating the benefits of our geometric-aware approaches. Our implementation builds upon the codebase of OT-CFM (Tong et al., 2024) and LightningDiT (Yao et al., 2025).

**Evaluation Metrics.** We evaluate all methods using standard generative modeling metrics computed on 50,000 generated samples. For quantitative assessment, we report Generative Fréchet Inception Distance (gFID) (Heusel et al., 2017) to measure the distributional distance between generated and real images, sFID (Nash et al., 2021), a variation of FID using spatial features, better captures spatial relationships and high-level structure in image distributions, Inception Score (IS) (Salimans et al., 2016) to evaluate sample quality and diversity, and Precision and Recall (Nichol & Dhariwal, 2021) to assess fidelity and coverage of the generated distribution.

| Space | rFID | | LPIPS | |
|---|---|---|---|---|
| | Baseline | Mean Projected | Baseline | Mean Projected |
| RGB | 0 | 0.05 (+0.05) | 0 | 0.02 (+0.02) |
| SD2-VAE | 0.71 | 1.14 (+0.43) | 0.13 | 0.15 (+0.02) |
| SD3-VAE | 0.20 | 0.40 (+0.20) | 0.06 | 0.08 (+0.02) |
| VMAE | 0.89 | 0.88 (-0.01) | 0.06 | 0.06 (+0.00) |
| DC-AE | 1.02 | 1.57 (+0.55) | 0.17 | 0.18 (+0.01) |

Table 1: Impact of hyperspherical projection on reconstruction quality across different representation spaces. Values show rFID and LPIPS metrics comparing original data with data projected onto hyperspheres using mean magnitude (Mean Projected). Changes from baseline are shown in parentheses.

## 4.2 ANALYSIS OF REPRESENTATION SPACES

Before presenting our main results, we validate our core hypothesis about the spherical nature of image data across different representation spaces. Table 1 shows the impact of projecting datasets onto hyperspheres with global average magnitude $\bar{s}$ while preserving directional components. We evaluate reconstruction quality using rFID (FID between original and projected data) and LPIPS metrics. The results confirm our hypothesis that across RGB space and multiple autoencoder latent spaces (SD2-VAE, SD3-VAE, VMAE, DC-AE), projecting data onto a hypersphere with average radius preserves most semantic information. Notably, the SD3-VAE latent space shows particularly robust behavior with only a 0.20 increase in rFID, while VMAE demonstrates near-perfect preservation with a negligible $-0.01$ change. Additionally, LPIPS maintains near-baseline performance across all representation spaces. This validates our assumption that directional information dominates semantic content across diverse representation spaces.

| Method | Source | Target | CIFAR10 | Class-Conditional ImageNet256 | | | | |
| --- | --- | --- | --- | --- | --- | --- | --- | --- |
| | | | gFID↓ | gFID↓ | sFID↓ | IS↑ | Precision↑ | Recall↑ |
| I-CFM | $\mathcal{N}$ | $\mathcal{D}$ | 4.29 | 5.56 | **7.42** | 199.25 | 0.7724 | 0.5124 |
| I-CFM (ours) | $\mathcal{N}$ | $\tilde{\mathcal{D}}$ | **4.10** | **5.01** | 7.89 | **231.70** | **0.8056** | **0.5432** |
| OT-CFM | $\mathcal{N}$ | $\mathcal{D}$ | 4.30 | 5.56 | 7.59 | 193.99 | 0.7656 | 0.5381 |
| SOT-CFM (ours) | $\mathcal{N}$ | $\mathcal{D}$ | **4.11** | **5.15** | 7.64 | **210.02** | **0.7839** | **0.5182** |
| SOT-CFM (ours) | $\mathcal{N}$ | $\tilde{\mathcal{D}}$ | 4.12 | 5.28 | **7.32** | 204.41 | 0.7742 | 0.5172 |
| SFM | $\mathcal{N}$ | $\mathcal{D}$ | *inapplicable* | | | | | |
| SFM (ours) | $\tilde{\mathcal{N}}$ | $\tilde{\mathcal{D}}$ | **3.79** | **6.22** | **10.84** | **218.88** | **0.7531** | **0.5140** |

Table 2: Generative performance comparison on unconditional CIFAR-10 and class-conditional ImageNet-256 (with CFG scale 2). $\mathcal{N}$ and $\tilde{\mathcal{N}}$ denote the Gaussian and spherically projected Gaussian distributions (which is nearly identical to uniform distribution on the sphere), respectively. $\mathcal{D}$ and $\tilde{\mathcal{D}}$ denote the original and spherically projected datasets, respectively. SFM is inherently designed for spherical manifolds and cannot be directly applied to Euclidean data. **Bold** values indicate the best performance within each method family.

## 4.3 QUANTITATIVE COMPARISON

We present our main experimental results in Table 2, comparing our proposed spherical methods against Euclidean baselines on the CIFAR-10 and ImageNet-256 datasets. The results validate that leveraging the inherent spherical geometry of image data leads to benefits in generative performance and enables the first successful application of manifold-based generative modeling to natural images.

**Utilizing Spherical Geometry.** Our first set of experiments confirms that projecting the data onto a hypersphere before training consistently improves standard Euclidean methods. Results for the spherically projected dataset $\tilde{\mathcal{D}}$ include post-processing magnitude adjustment. Spherical projection consistently improves I-CFM performance, reducing the FID from 4.29 to 4.10 on CIFAR-10 and from 5.56 to 5.01 on ImageNet-256, which corresponds to a substantial 10% improvement on this more challenging benchmark.

Furthermore, our SOT-CFM, which is designed to focus on semantically meaningful directional information by using an angular transport cost, outperforms its direct baseline. On CIFAR-10, SOT-CFM achieves an gFID of 4.11, a notable improvement over OT-CFM's 4.30. Interestingly, we observe that combining spherical projection with SOT-CFM yields marginal performance degradation, with gFID slightly increasing from 4.11 to 4.12 on CIFAR-10 and from 5.15 to 5.28 on ImageNet-256. Collectively, these results provide strong evidence for our claim that leveraging the directional components of image data is an effective path to better generation.

**Spherical Flow Matching.** Our SFM method, which defines source and target distributions as well as flow paths on the hyperspherical manifold, achieves the strongest performance on CIFAR-10 with an gFID of 3.79 and outperforming all other variants.

However, on ImageNet-256, SFM does not surpass existing methods, achieving an gFID of 6.22. We attribute this performance gap primarily to our current implementation being a foundational Riemannian-based approach that has not yet benefited from the extensive engineering optimizations typically applied to such challenging scenarios. Although SFM does not achieve state-of-the-art performance at this stage, it demonstrates meaningful results and represents a successful application of Riemannian-based methods to natural image generation. To our knowledge, this is the first generative model to operate within a Riemannian manifold for image generation task. This breakthrough makes sophisticated geometric tools accessible in the natural image domain, serving as an important bridge between theoretical manifold-based methods and practical large-scale applications.

## 5 RELATED WORKS

**Flow Matching.** Flow Matching learns vector fields that transport noise to data distributions. Unlike Continuous Normalizing Flows (CNFs) (Chen et al., 2018; Grathwohl et al., 2018), which suffer

from expensive trace computations, Flow Matching (Lipman et al., 2023; Liu et al., 2022; Albergo et al., 2023) uses a simulation-free objective to directly regress onto the conditional vector field, improving scalability.

Building on these foundations, Flow Matching has been extended to support diverse probability paths (Gat et al., 2024; Stark et al., 2024; Cheng et al., 2025; Kapusniak et al., 2024). The method has found applications across multiple domains, including image generation (Esser et al., 2024; Dao et al., 2023; Ren et al., 2024), audio synthesis (Guan et al., 2024; Liu et al., 2023; Prajwal et al., 2024), video generation (Jin et al., 2025; Polyak et al., 2024), molecular modeling (Dunn & Koes, 2024; Song et al., 2023), and text generation (Hu et al., 2024).

**Geometry-Aware Generative Modeling.** Incorporating geometric priors has proven effective for data on known manifolds. Riemannian Continuous Normalizing Flows (Mathieu & Nickel, 2020) first integrated differential geometry into flow-based models via manifold-aware ODEs. Similarly, Riemannian Score-Based (Bortoli et al., 2022) and Diffusion Models (Huang et al., 2022) extended diffusion processes to Riemannian manifolds, improving performance on structured data. More recently, Riemannian Flow Matching (RFM) (Chen & Lipman, 2024) enabled simulation-free training on manifolds, and FlowMM (Miller et al., 2024) extended it with group-equivariance for efficient crystal generation.

**Optimal Transport in Generative Modeling.** Optimal Transport (OT) provides a theoretical foundations for generative models, notably used in Wasserstein GANs (Arjovsky et al., 2017) for stable adversarial training. In Flow Matching, OT offers principled pairings between source and target points (Tong et al., 2024; Pooladian et al., 2023), leading to shorter transport paths with improved efficiency. However, existing OT methods often rely on Euclidean metrics which may not be optimal for high-dimensional image data where directional similarity could be more meaningful. Scalable algorithms like Sinkhorn iterations (Cuturi, 2013) and progressive solvers (Kassraie et al., 2024) have made OT practical for large-scale tasks. Further refinements address challenges in minibatch conditional settings through class-aware penalties (Cheng & Schwing, 2025) or partial optimal transport (Nguyen et al., 2022), while others learn the coupling strategy directly from data (Lin et al., 2025).

**Autoencoders in the Latent Diffusion Framework.** Latent diffusion models rely on an autoencoder to compress images into a meaningful latent space. The Autoencoder-KL from Stable Diffusion (Rombach et al., 2022) established this approach, which was later scaled up in SDXL (Podell et al., 2023) and Stable Diffusion v3 (Esser et al., 2024) for improved perceptual fidelity and conditioning strategies. Beyond these, alternative encoders have been proposed to enhance latent representations. VA-VAE (Yao et al., 2025) addresses the reconstruction-generation trade-off in latent diffusion by aligning the VAE latent space with a pretrained vision foundation model. MAEtok (Chen et al., 2025) and VMAE (Lee et al., 2025) combined latent diffusion with masked autoencoding, improving latent space quality. In parallel, deep compression autoencoders (DC-AE) (Chen et al., 2024) were designed to reduce latent dimensionality while maintaining fidelity.

## 6 CONCLUSION

In this work, we investigate the role of geometric structure in image generation by identifying a spherical property of natural images through directional decomposition, where semantic information is largely encoded in directions while scalar components can be approximated by dataset averages. Building on this observation, we introduce geometry-aware flow matching methods for images, SOT-CFM and SFM, which employ angular metrics and geodesic dynamics on the hypersphere to produce geometrically consistent generative paths. Experiments on CIFAR-10 and ImageNet-256 show that these spherical approaches achieve comparable or better performance than Euclidean baselines across multiple metrics and representation spaces, including RGB and modern autoencoder latents, indicating their broad applicability in image domains. Overall, our results establish that leveraging the intrinsic geometric structure of image data provides a principled and effective perspective for generative modeling. This work opens the door to geometry-aware approaches in image generation, demonstrating that even when explicit geometric priors are unknown, careful analysis can reveal exploitable manifold structure that improves upon standard Euclidean methods.

## REPRODUCIBILITY STATEMENT

To ensure the reproducibility of our work, we have taken extensive measures to provide all essential implementation details and experimental configurations. Specifically, Section 4.1 presents the complete experimental setup, covering datasets, model architectures, training procedures, evaluation metrics, as well as all hyperparameter and architectural choices that were adopted in our experiments. In addition, the source code, together with detailed instructions for replication, will be made publicly available upon publication.

## USE OF LARGE LANGUAGE MODELS (LLMS)

We used LLMs only as a general-purpose assist tool for grammar checking, sentence refinement, LaTeX syntax validation, table formatting, and as a code assistant for debugging and formatting scripts. They did not contribute to the research ideation, methodological design, analysis, or writing of substantive scientific content.

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
