# OpenReview forum: "Geometry-Aware Image Flow Matching"
_ICLR.cc/2026/Conference — ICLR 2026 Conference Withdrawn Submission_

### Official Review · Reviewer_1qdM · 2025-10-24

**Soundness:** 1
**Presentation:** 2
**Contribution:** 2
**Rating:** 2
**Confidence:** 4

**Summary:**

The paper proposes a new approach to the problem of image generation through flow matching, motivated by the goal of respecting the intrinsic geometric structure of images. First, the authors experimentally observe that the semantic information of images can be encoded in the directional component of the image vectors, hence as different points on a hypersphere, once the scalar component is set to be constant for all images in the dataset. Motivated by this, the work proposes two hypersphere-dependent flow matching methods for generating images and validates them in the generation of images from two different real-world datasets, comparing them to Euclidean flow matching methods.

**Strengths:**

- The work addresses a new and challenging problem providing a novel and interesting solution.
- The paper is well organized in the structure and it reads well. The motivations, objective, and contributions are well explained and the necessary background knowledge is clearly and extensively provided.

**Weaknesses:**

The paper makes a strong claim of addressing the data manifold hypothesis by suggesting that most of the semantic information in images is encoded in their directional component, thereby justifying an encoding on the surface of a hypersphere. However, this claim is not sufficiently validated in practice. Specifically:
- The robustness analysis presented in Figure 1 is conducted on a single dataset; evaluating on additional datasets would be important to assess generality and robustness.
- In the image generation experiments, imposing the hypersphere constraint does not consistently improve performance -- in some cases, it even degrades the results according to the reported metrics.
- The necessity of an additional Magnitude Prediction network to achieve faithful image reconstruction suggests that the proposed approach alone may be insufficient.

Overall, while the contribution is interesting, the paper’s central message would benefit from being reframed. Rather than claiming to fully capture the true geometric structure of the data, the work could more appropriately be presented for example as an attempt to reduce the dimensionality of the space in which flow matching operates, by approximating image data through their projection onto a lower-dimensional hyperspherical representation.

**Questions:**

These are my main questions and observations on points to be fixed:

**General concerns**
- The paper makes an ambitious claim that appears to address the data manifold hypothesis directly. I would recommend restructuring the stated goals to make them more realistic and focused - for instance, framing the objective as reducing the space in which flow matching operates by proposing an approach that considers an approximation of image data by projecting them onto a lower-dimensional space for generation.
- It seems that the claim is not strongly supported by the analysis/experiments. In particular: (1) the robustness analysis shown in Figure 1 is conducted on only one dataset, and it would likely be more appropriate to include additional datasets for generality test, and (2) based on the results, imposing the hypersphere constraint does not always improve performance, as evidenced by the degradation observed with the SOT-CFM loss.
- Following from the previous point, could you please elaborate more on what "slight loss" means in Section 3.4, second line? It would be helpful to have a quantitative analysis of this to better understand to which extend the  Magnitude Refinement Network is essential.

**Experiments**
- In Table 2, the reported values are not properly interpretable without a corresponding confidence bound +/-std. Would it be possible to have such interval over the executed runs?
- In Table 2, the bolding of the number is done incorrectly in the last column (Recall), for the second row group, as OT-CFM appears to be the best one.
- It would be interesting to visually observe what some generated images look like from the considered datasets, and also to be able to compare how the images look with and without using the additional Magnitude Refinement Network, as a sort of ablation study to further address how the hypersphere constraint affects the quality of the images.
- How would you explain that the performance of SOT-CFM degradates by applying spherical projection?

**Paper structure and writing**
- Figure 2 is never cited.
- Some acronyms are introduced early in the paper and explained only later (e.g. rFID and LPIPS introduced in Section 3.1 but explained in the Experiment section), or never explicitly explained (e.g. LDM in Section 3.1).

---

### Official Review · Reviewer_sxMs · 2025-11-01

**Soundness:** 3
**Presentation:** 3
**Contribution:** 3
**Rating:** 4
**Confidence:** 3

**Summary:**

The authors analyse the geometry of natural images and find that when considered as vectors, both in pixel-space and the latent space of autoencoders, the norm is practically fixed and can be approximated by the dataset’s average norm. Therefore, we can consider natural images as living on the hyper-sphere manifold. The authors then exploit this via applying Riemannian flow matching on the hyper-sphere for image generation. This aims to simplify the learning problem as the degrees of freedom of the problem is reduced to the manifold instead of the ambient Euclidean space. They also introduce an optimal transport-based coupling specialised to the hyper-sphere to improve performance. Through empirical evaluations on CIFAR-10 and ImageNet, they demonstrate competitive performance with baselines.

**Strengths:**

* I like the central idea of exploiting the geometric structure of natural images in a principled way, in particular, the novelty of bringing in work on manifold-based generative modelling to this problem domain.

* The development of spherical flow matching and OT-based coupling is carried out in a principled and well-motivated manner, although the choices here are fairly straight forward.

* There is a good range of metrics in Table 2 evaluating performance of image generation, going beyond just FID.

**Weaknesses:**

* It would be informative to see ablations on the impact of the choice of radius on training.

* It appears that the related work section neglects to mention the whole area of work investigating the structure of natural images, such as [1, 2] among many others. Also, this work [3] should be included in the references due to the focus on modelling distributions on the hyper-sphere.

* In Table 1, it is unclear that dataset is being considered - i.e. are you looking at ImageNet or CIFAR-10?

* In the empirical results, I cannot see what the sampling hyper-parameters are. For example, are the number of function evaluations for the baselines and SFM kept the same?

* The paper claims that learning only the directional component does simplify the learning problem as only trajectories on the manifold needs to be considered, however, this is only removing a single degree of freedom from the high dimensional image space, which isn't a huge simplification. It would be good to further explore theoretically what benefits and empirically that the hyper-sphere model can bring.

* What is the additional cost of training the Magnitude Refinement Network as well as the performance it has during sampling? There should be ablations for this to understand how much the additional compute overhead brings.

[1] On the Local Behavior of Spaces of Natural Images (2008)

[2] The statistics of natural images (1994)

[3] Normalizing Flows on Tori and Spheres (2020)

**Questions:**

* Do you have FID results for the experiment in Table 2 as well.
* Have you investigate the geometric structure of submanifold in the data - for instance, looking at the geometry of a given class in ImageNet and whether this also can be approximated by a hyper-sphere.
* The paper appears to be missing the appendix.

---

### Official Review · Reviewer_8BpM · 2025-11-01

**Soundness:** 2
**Presentation:** 2
**Contribution:** 2
**Rating:** 2
**Confidence:** 2

**Summary:**

The paper studies the geometry of image data through a directional plus scalar decomposition, and claims that most semantic information is in the direction, while the magnitude can be set near a global average. From this diagnosis the paper proposes two geometry aware flow matching methods. First, SOT CFM uses an angular cost inside minibatch optimal transport. Second, SFM projects source and target to a sphere and trains the vector field along geodesic paths given by slerp, with an extra lightweight magnitude refinement module. Experiments on CIFAR 10 and class conditional ImageNet 256 compare I CFM, OT CFM, SOT CFM, and SFM. On CIFAR 10, SFM reports the best gFID, for example gFID 3.79, while on ImageNet 256 SFM is worse than SOT CFM, for example gFID 6.22 for SFM versus 5.15 for SOT CFM. Figure 1 on page 4 shows that projecting to a fixed radius changes rFID only modestly over a wide band in RGB and SD 3 VAE space, Table 1 on page 7 summarizes rFID and LPIPS deltas across several latent spaces, and Table 2 on page 8 reports the main generative metrics.

**Strengths:**

### **Strengths**

- Clear problem framing and algorithms. The paper explains the directional plus scalar view, and gives concrete spherical variants of CFM with precise paths and tangent space training. The diagram on page 5 helps.
- Evidence that angular coupling can help within the CFM family. On CIFAR 10, SOT CFM improves over OT CFM on gFID.
- A reasonably thorough first pass at applying manifold aware training to images, with code release promised.

**Weaknesses:**

### **Weaknessess**

- Though the others show some experimental improvement, the results are somewhat marginal and evaluated on a limited scope. As such, I believe the practical contribution of this work is marginal. Moreover, the spherical method that most reflects the goal of the paper (SFM), though being best on CIFAR 10 is worse on ImageNet 256 relative to SOT CFM.
- Using angular distance for coupling and constraining flows to the sphere follows directly from standard geometry, and the paper does not provide a new theoretical insight that changes our understanding of CFM itself.
- The paper mainly measures rFID and LPIPS between original data and data reconstructed after spherical projection. These measures indicate distribution and perceptual similarity, but simple tests such as classification accuracy under magnitude changes, or accuracy on tasks where brightness matters, would be more decisive. I hence believe the theoretical claims/assumptions could have been ablated much more strongly to improve the work.


In general I think this work has potential but just in need of one more round of experimental validation before acceptance.

**Questions:**

- Can you report top one accuracy of a standard ImageNet classifier on original images versus their projected reconstructions across a sweep of radii, and also the relative drop under stronger magnitude perturbations
- What is the empirical distribution of pairwise angles before and after projection in RGB and in each latent space, and how does this affect SOT CFM coupling quality at realistic batch sizes
- How often do slerp paths encounter near zero or near pi angles during training, and does this matter in practice?
- Can you integrate magnitude prediction into the flow objective, rather than as a separate regressor, and measure its effect on precision and recall on ImageNet 256
- Why does combining spherical projection with SOT CFM hurt slightly on ImageNet 256 in Table 2, even though the direction based cost seems aligned with the thesis
- How sensitive are the methods to the chosen radius during training, beyond the reconstruction curves in Figure 1, for example convergence speed and final gFID across different radii and seeds

---

### Official Review · Reviewer_qWrs · 2025-11-03

**Soundness:** 3
**Presentation:** 3
**Contribution:** 2
**Rating:** 2
**Confidence:** 3

**Summary:**

The paper proposes geometry-aware flow matching for image datasets. The authors empirically show that the generative modeling of natural images mainly involves modeling the directional component, while the scalar component has minimal impact. Based on this finding, the authors propose to project the image data into the surface of a hypersphere with a fixed radius (set to the average magnitude of the data). They propose to train geometry-aware flow matching on the surface of the sphere. Through numerical experiments, the authors show improvement in generative performance with their approach compared to the baselines.

**Strengths:**

Motivation and presentation of the method are clear. It is interesting to see that the directional components of an image are the main features, and the scalar component can be ignored. The experiments do show improvement with the proposed method compared to OTCFM and ICFM.

**Weaknesses:**

Major:

Weak baselines - The authors compare their method to only manifold-unaware generative models. Recent methods, such as Metric-FM (Kapusniak et al. 2024), Latent-CFM (Samaddar et al. 2025) propose modifications to the flow matching training to incorporate structures of the data and the underlying geometry into the learned flow. Comparing the current method to existing geometry-aware flow matching approaches will increase the thoroughness of the evaluation.

Other:

Use of LPIPS in the loss - The authors have used a regularization based on LPIPS in their loss function. However, the other baselines do not have the same fidelity regularizers. Therefore, it is unclear how much the performance gain is due to the spherical projection.  Can the authors present results of ablation studies without the regularizer?

**Questions:**

1.  In Fig.1, how was the range for the scalar magnitudes chosen?

2. In line 292, the authors claim that they exploit intrinsic geometric structures of the data. However, current work projects the data into a hypersphere and then exploits the structure of the sphere. Can the authors explain what they mean by exploiting the intrinsic structure of the data?

3. Can the authors report NFEs for their approach and the baselines in Table 2? Additionally, it is unclear what strategies (solvers) were employed during inference.

---

### Note · Authors · 2025-11-12

I have read and agree with the venue's withdrawal policy on behalf of myself and my co-authors.